# Digitally enabled aged care and neurological rehabilitation to enhance outcomes with Activity and MObility UsiNg Technology (AMOUNT) in Australia: A randomised controlled trial

**Leanne Hassett**[1,2], **Maayken van den Berg**[3,4], **Richard I. Lindley**[5], **Maria Crotty**[3], **Annie McCluskey**[2,6], **Hidde P. van der Ploeg**[7,8], **Stuart T. Smith**[9], **Karl Schurr**[6], **Kirsten Howard**[8], **Maree L. Hackett**[10,11], **Maggie Killington**[3], **Bert Bongers**[12], **Leanne Togher**[2], **Daniel Treacy**[1,13], **Simone Dorsch**[6,14,15], **Siobhan Wong**[1,16], **Katharine Scrivener**[6,17], **Sakina Chagpar**[1], **Heather Weber**[3], **Marina Pinheiro**[1,2], **Stephane Heritier**[18], **Catherine Sherrington**[1] *

1 Institute for Musculoskeletal Health, Faculty of Medicine and Health, University of Sydney, Sydney, New South Wales, Australia, 2 School of Health Sciences, Faculty of Medicine and Health, University of Sydney, Sydney, New South Wales, Australia, 3 Rehabilitation, Aged and Extended Care, College of Medicine and Public Health, Flinders University, Adelaide, South Australia, Australia, 4 Clinical Rehabilitation, College of Nursing and Health Sciences, Flinders University, Adelaide, South Australia, Australia, 5 Westmead Clinical School, Faculty of Medicine and Health, University of Sydney, Sydney, New South Wales, Australia, 6 StrokeEd Collaboration, Sydney, New South Wales, Australia, 7 Department of Public & Occupational Health, Amsterdam Public Health Research Institute, Amsterdam UMC, Vrije Universiteit Amsterdam, Amsterdam, Netherlands, 8 School of Public Health, Faculty of Medicine and Health, University of Sydney, Sydney, New South Wales, Australia, 9 School of Health and Human Sciences, Southern Cross University, Coffs Harbour, New South Wales, Australia, 10 The George Institute for Global Health, Faculty of Medicine, University of New South Wales, Sydney, New South Wales, Australia, 11 Faculty of Health and Wellbeing, University of Central Lancashire, Preston, United Kingdom, 12 Faculty of Design, Architecture and Building, University of Technology Sydney, Sydney, New South Wales, Australia, 13 Physiotherapy Department, Prince of Wales Hospital, South Eastern Sydney Local Health District, Sydney, New South Wales, Australia, 14 Physiotherapy Department and Department of Aged Care and Rehabilitation, Bankstown-Lidcombe Hospital, South Western Sydney Local Health District, Sydney, New South Wales, Australia, 15 School of Physiotherapy, Faculty of Health Sciences, Australian Catholic University, Sydney, New South Wales, Australia, 16 Physiotherapy Department and Brain Injury Rehabilitation Unit, Liverpool Hospital, South Western Sydney Local Health District, Sydney, New South Wales, Australia, 17 Faculty of Medicine and Health Sciences, Macquarie University, Sydney, New South Wales, Australia, 18 Department of Epidemiology and Preventive Medicine, Faculty of Medicine, Nursing and Health Sciences, Monash University, Melbourne, Victoria, Australia

* cathie.sherrington@sydney.edu.au

## Abstract

### Background

Digitally enabled rehabilitation may lead to better outcomes but has not been tested in large pragmatic trials. We aimed to evaluate a tailored prescription of affordable digital devices in addition to usual care for people with mobility limitations admitted to aged care and neurological rehabilitation.

**Data Availability Statement:** The data underlying the results presented in the study are available from the University of Sydney's open access institutional repository, Sydney eScholarship repository: https://ses.library.usyd.edu.au/handle/2123/21698.

**Funding:** This work was supported by an Australian National Health and Medical Research Council Project Grant (APP1063751). CS receives salary funding from an Australian National Health and Medical Research Council Fellowship. No funding bodies had any role in the study design, data collection and analysis, decision to publish, or preparation of this manuscript.

**Competing interests:** I have read the journal's policy and the authors of this manuscript have the following competing interests: MH is in receipt of a National Health and Medical Research Council of Australia Career Development Fellowship Level 2, APP1141328.

**Abbreviations:** AMOUNT, Activity and MObility UsiNg Technology; IPEQ, Incidental and Planned Exercise Questionnaire; SPPB, Short Physical Performance Battery.

## Methods and findings

We conducted a pragmatic, outcome-assessor-blinded, parallel-group randomised trial in 3 Australian hospitals in Sydney and Adelaide recruiting adults 18 to 101 years old with mobility limitations undertaking aged care and neurological inpatient rehabilitation. Both the intervention and control groups received usual multidisciplinary inpatient and post-hospital rehabilitation care as determined by the treating rehabilitation clinicians. In addition to usual care, the intervention group used devices to target mobility and physical activity problems, individually prescribed by a physiotherapist according to an intervention protocol, including virtual reality video games, activity monitors, and handheld computer devices for 6 months in hospital and at home. Co-primary outcomes were mobility (performance-based Short Physical Performance Battery [SPPB]; continuous version; range 0 to 3; higher score indicates better mobility) and upright time as a proxy measure of physical activity (proportion of the day upright measured with activPAL) at 6 months. The dataset was analysed using intention-to-treat principles. The trial was prospectively registered with the Australian New Zealand Clinical Trials Registry (ACTRN12614000936628). Between 22 September 2014 and 10 November 2016, 300 patients (mean age 74 years, SD 14; 50% female; 54% neurological condition causing activity limitation) were randomly assigned to intervention ($n = 149$) or control ($n = 151$) using a secure online database (REDCap) to achieve allocation concealment. Six-month assessments were completed by 258 participants (129 intervention, 129 control). Intervention participants received on average 12 (SD 11) supervised inpatient sessions using 4 (SD 1) different devices and 15 (SD 5) physiotherapy contacts supporting device use after hospital discharge. Changes in mobility scores were higher in the intervention group compared to the control group from baseline (SPPB [continuous, 0–3] mean [SD]: intervention group, 1.5 [0.7]; control group, 1.5 [0.8]) to 6 months (SPPB [continuous, 0–3] mean [SD]: intervention group, 2.3 [0.6]; control group, 2.1 [0.8]; mean between-group difference 0.2 points, 95% CI 0.1 to 0.3; $p = 0.006$). However, there was no evidence of a difference between groups for upright time at 6 months (mean [SD] proportion of the day spent upright at 6 months: intervention group, 18.2 [9.8]; control group, 18.4 [10.2]; mean between-group difference −0.2, 95% CI −2.7 to 2.3; $p = 0.87$). Scores were higher in the intervention group compared to the control group across most secondary mobility outcomes, but there was no evidence of a difference between groups for most other secondary outcomes including self-reported balance confidence and quality of life. No adverse events were reported in the intervention group. Thirteen participants died while in the trial (intervention group: 9; control group: 4) due to unrelated causes, and there was no evidence of a difference between groups in fall rates (unadjusted incidence rate ratio 1.19, 95% CI 0.78 to 1.83; $p = 0.43$). Study limitations include 15%–19% loss to follow-up at 6 months on the co-primary outcomes, as anticipated; the number of secondary outcome measures in our trial, which may increase the risk of a type I error; and potential low statistical power to demonstrate significant between-group differences on important secondary patient-reported outcomes.

## Conclusions

In this study, we observed improved mobility in people with a wide range of health conditions making use of digitally enabled rehabilitation, whereas time spent upright was not impacted.

## Trial registration

The trial was prospectively registered with the Australian New Zealand Clinical Trials Register; ACTRN12614000936628

---

Author summary

### Why was this study done?

- A higher dose of therapy in physical rehabilitation is associated with better outcomes; however, current rehabilitation models deliver low therapy doses.

- Use of digital devices such as virtual reality video games, activity monitors, and hand-held computer devices can be enjoyable, provide feedback on performance, and may enable a greater dose of task-specific therapy to improve outcomes.

- Current evidence is yet to confidently confirm the effects of rehabilitation using digital devices in addition to usual rehabilitation care on mobility tasks such as walking and other important outcomes such as quality of life.

### What did the researchers do and find?

- In a pragmatic, outcome-assessor-blinded randomised controlled trial, 300 people with walking difficulties (age 72 ± 16 years, 50% female) received usual multidisciplinary inpatient and post-hospital aged care and neurological rehabilitation alone, or in addition used a range of affordable devices such as virtual reality video games, activity monitors, and handheld devices to target mobility and physical activity, as individually prescribed by a physiotherapist for 6 months.

- On average participants in the intervention group used 4 ± 1 devices in the inpatient setting and 2 ± 1 devices in the post-hospital setting. This approach was feasible and enjoyed, and demonstrated it could be provided across care settings including the post-hospital setting with mostly remote support.

- Clinically important improvement was seen in mobility at 3 weeks and 6 months after baseline, but this was not accompanied by greater time spent upright.

- No adverse events were reported by participants whilst undertaking rehabilitation using digital devices, and there was no difference in the rate of falls between groups.

### What do these findings mean?

- Digitally enabled rehabilitation using a range of devices prescribed by a physiotherapist to target a range of mobility limitations across care settings for adults with mixed health conditions can improve mobility but not time spent upright.

- These results need to be interpreted in light of study limitations including a 15%–19% loss to follow-up at 6 months on the co-primary outcomes.

- Future models of rehabilitation should investigate incorporating digital devices to enhance inpatient and post-hospital rehabilitation, but prescription should ensure quality and quantity of practice.

## Introduction

Over 20% of the world population will be >60 years of age by 2050 [1]. Many will need accessible and affordable rehabilitation to reduce costly limitations in function from neurological and musculoskeletal health conditions [2] as well as decline from aging and inactivity [3]. Physical rehabilitation should contain intensive, repetitive task-specific exercises to improve outcomes [4–7]. Virtual reality video games, activity monitors, and handheld computer devices are accessible, affordable, and enjoyable [8], and together can provide a digitally enabled rehabilitation environment by providing more opportunity and greater motivation to increase task-specific practice in hospital [9] and in the home setting [10]. However, evidence of their impact on outcomes is limited and focused on stroke rehabilitation [11,12]. A systematic review of virtual reality interventions in people after stroke (72 studies) demonstrated a moderate effect on balance, but no effect on walking speed or global motor function when delivered as an adjunct to usual rehabilitation [11]. However, the quality of evidence was rated as low for nearly all outcomes, and all but 1 study tested a single virtual reality system. A feasibility trial conducted by our team in people undertaking inpatient aged care and neurological rehabilitation ($n$ = 58) provided an additional dose of rehabilitation for 2 weeks using a range of low-cost video games and activity monitors [13]. The intervention was feasible, safe, and enjoyable, and enabled a higher dose of exercise and improved balance but not overall mobility. This promising intervention, after refinement, required rigorous evaluation.

The primary aim of the Activity and MObility UsiNg Technology (AMOUNT) trial was to test the effectiveness of tailored prescription of affordable devices to improve mobility and physical activity in people with mobility limitations undertaking aged care and neurological rehabilitation. The devices were prescribed in addition to usual care and compared to usual care alone.

## Methods

### Design

AMOUNT was a pragmatic, assessor-blinded, multicentre superiority randomised controlled trial with 2 parallel groups and included a nested economic analysis (presented separately) and a qualitative study [14].

### Sites, staff, and participants

There were 3 trial sites across Australia. Two were in metropolitan hospitals in Sydney in New South Wales (Site 1: 20-bed stroke and 20-bed aged care rehabilitation wards; Site 2: 16-bed brain injury rehabilitation unit), and 1 was in Adelaide in South Australia (Site 3: 30-bed geriatric evaluation and management ward and 40- and 20-bed general rehabilitation wards). Research physiotherapists recruited participants, conducted baseline assessments, randomised participants, and delivered the intervention; all were experienced physiotherapists and received training in trial processes, as well as in the digital intervention to be delivered.

Consecutive patients admitted to the units who met the following criteria were invited to participate: ≥18 years old; reduced mobility (Short Physical Performance Battery [SPPB] score < 12) [15] with clinician-assessed capacity for improvement (based on the usual care physiotherapists' clinical experience and their assessment and treatment experience with the patient); life expectancy > 12 months; anticipated length of stay ≥ 10 days from randomisation; and able to maintain a standing position (with assistance of 1 person if necessary). Patients were excluded if they had any of the following: cognitive impairment likely to interfere with device use; insufficient English language skills with no available interpreter; inadequate vision to use devices; medical condition(s) precluding exercise; no interest in using devices; anticipated discharge to high care residential facility (nursing home); or discharge location too distant for follow-up.

## Randomisation

A staff member external to the trial prepared the randomisation schedule using randomly permuted block sizes of 2, 4, and 6 and incorporating stratification for study site and health condition (whether or not the person had a neurological health condition affecting mobility). Following written informed consent and baseline assessment, research staff completed web-based randomisation (allocation concealment) to determine group allocation.

## Intervention

Both the intervention and control groups received usual rehabilitation care, which was determined by the treating clinicians and included assessment and prescription of a series of repetitive exercises by the physiotherapist, tailored management by the multidisciplinary team, and a fall prevention brochure [16] (see Table 1). In addition, the intervention group was prescribed 30 to 60 minutes of digitally enabled rehabilitation 5 days per week in hospital and post-discharge, defined as rehabilitation using digital devices (e.g., virtual reality, wearables, and tablet and smartphone applications), with remote monitoring and communication post-discharge. The intervention group was prescribed exercises using virtual reality video games, activity monitors, and handheld computer devices to enhance mobility and physical activity. The exercises and devices were individually prescribed by a trial physiotherapist according to an intervention protocol that matched different task-specific exercises on different devices to common mobility limitations. The physiotherapist also considered participant impairments (e.g., upper limb weakness, hemianopia) and contextual factors such as participant goals, device preferences, and the home environment. Included devices were purchased by the research team or constructed for less than US$3,700 each. Participants could use any number of devices as guided by the physiotherapist. Devices were loaned to participants to use at home and were progressed or changed as required. For further details of usual care and the additional intervention using digital devices, see Table 1 and S1 Text, and the published protocol [17].

## Outcome measures

Face-to-face outcome assessments were conducted at 3 weeks and 6 months after randomisation and by mail or telephone at 12 weeks after randomisation. Outcome assessors were registered health professionals trained in conducting the outcome assessments, external to the clinical sites, and blinded to group allocation. The face-to-face assessments were conducted in the hospital if the participant was still an inpatient, or at the post-hospital-discharge destination (e.g., home, transitional living unit). Prior to the outcome assessor completing the

**Table 1. Intervention description using the template for intervention description and replication (TIDieR) checklist.**

| Checklist item | Intervention group | | Control group | |
|---|---|---|---|---|
| | Inpatient setting | Post-hospital setting | Inpatient setting | Post-hospital setting |
| **Brief name** | Digitally enabled rehabilitation in addition to usual care. | | Usual care. | |
| **Why** | Digital devices potentially provide an affordable way to increase the dose of practice for better rehabilitation outcomes. Devices such as virtual reality video games, activity monitors, and handheld computer devices enhance enjoyment of exercise and provide feedback for motor relearning. | | Pragmatic trial design. | |
| **What** | | | | |
| Materials for therapists | A detailed intervention protocol that matched mobility limitations with different devices and games/exercises within those devices. Training in health coaching by an external provider or previously trained therapists. Research managers provided ongoing training on the use of the devices, clinical reasoning, and health coaching. | | Clinical therapists were provided with information on the trial protocol and asked not to use devices to improve mobility or physical activity as part of their usual care intervention. | |
| Materials for participants | Participants were (1) provided with a fall prevention brochure on discharge from hospital [16]; (2) loaned devices for the duration of the trial; (3) provided with trial-developed practice sheets and information sheets on how to use the different devices; (4) prescribed mobility exercises and/or physical activity using devices in addition to usual care. Recreational devices: Nintendo Wii (Nintendo, Kyoto, Japan); Xbox Kinect (Microsoft, Redmond, Washington, US); Fitbit Zip, One, and Alta (Fitbit, San Francisco, California, US); Garmin Vivofit (Garmin, Olathe, Kansas, US); Runkeeper mobile phone application (FitnessKeeper, Boston, Massachusetts, US). Rehabilitation devices: Humac Balance System (CSMi Solutions, Stoughton, Massachusetts, US); Fysiogaming (Doctor Kinetic, Amsterdam, the Netherlands). Investigator-developed devices: Stepping Tiles (University of Technology Sydney, Sydney, Australia); T-Rex iPad exercise application (Repatriation General Hospital, Adelaide and Sydney, Australia); AMOUNT iPad exercise application (University of Sydney, Sydney, Australia); Walk Forward iPhone application (The George Institute for Global Health and Telstra Health, Sydney, Australia). | | Participants were (1) provided with a fall prevention brochure on discharge from hospital [16]; (2) provided with inpatient usual care at the 3 study sites involving assessment and prescription of a series of repetitive exercises (e.g., practice of standing up or stepping); (3) referred to usual outpatient therapy as clinically required. Usual care also included assessment and tailored management by medical specialists, nurses, occupational therapists, speech pathologists, social workers, nutritionists, orthoptists, and other health professionals as required. | |
| **Who provided** | Physiotherapists employed on the trial. | | Physiotherapists employed at the study site hospitals. | No intervention or physiotherapists employed at the study site hospitals or private physiotherapists. |
| **How** | Face-to-face sessions. | Face-to-face and remote sessions following a health coaching model. | A mix of one-on-one, semi-supervised, independent, and group-based sessions. | |
| **Where** | Inpatient rehabilitation gym. | Remotely by phone/email/video conferencing or in person at the participant's discharge destination (home, transitional living unit, residential care). | Inpatient rehabilitation gym. | No intervention or outpatient rehabilitation gym, at the participant's discharge destination or in the community. |
| **When and how much** | ≥5 times per week for ≥30 minutes per session with physiotherapy supervision or monitoring. | ≥5 times per week for ≥30 minutes per session independently or with carer support. Research physiotherapists provided support using health coaching model every 1–2 weeks depending on participant needs and preferences. | Participants were seen as required by their treating physiotherapist: typically, ≥1 session per day Monday to Friday (and weekends for 1 site). | Participants who required ongoing physiotherapy were seen by outpatient/domiciliary physiotherapy services as required. |
| **Tailoring** | The intervention was tailored for each participant to address current mobility limitations and physical inactivity, considering participant goals, device preferences, and contextual factors (e.g., home environment). | | Determined by treating physiotherapist. | |

*(Continued)*

**Table 1.** (Continued)

| Checklist item | Intervention group | | Control group | |
| --- | --- | --- | --- | --- |
| | **Inpatient setting** | **Post-hospital setting** | **Inpatient setting** | **Post-hospital setting** |
| **Modifications** | As planned, the intervention protocol was modified during the trial; version 2 (published 14 October 2015) and version 3 (published 23 February 2016). Modifications included adding new games (e.g., Game Trainer for Nintendo Wii), a new iPhone application (Walk Forward), and upgrades of devices (e.g., software updates and rollout of a home-based version for Fysiogaming). Health coaching was initially prescribed weekly but changed within the first 6 months of the trial to 'as required' with a recommendation of weekly initially, reducing the frequency over time if the participant was managing well. This was modified due to experience in the trial and matched the tailored nature of the intervention (see S2 Text). | | Not applicable. | |
| **Trial fidelity** | Fidelity checking by site research managers (LH and MvdB) entailed observation of intervention sessions (inpatient and community), review of intervention data sheets with feedback/discussion, site weekly/fortnightly team meetings, combined-site quarterly meetings with case studies, practical sessions with devices, review of intervention protocol, and regular phone meetings between site research managers. | | Clinical practice sheets were collected from staff at the 2 sites in New South Wales (where it was usual practice for therapists to provide practice sheets) to assess usual physiotherapy care. Participants were questioned regarding their device use at the time of hospital discharge, and at the end of the trial intervention. | |

6-month assessment, the intervention devices were removed from participant homes and participants were reminded not to discuss their trial involvement with the assessor.

**Primary outcomes.** The co-primary outcomes were mobility and physical activity (upright time) 6 months after randomisation. Mobility is a broad term that is defined as the ability to move around and change positions, such as to stand up from sitting and to walk. Mobility was assessed with the performance-based SPPB (continuous version), also known as the lower extremity continuous summary performance score, which uses actual time taken to complete mobility tasks [18]. Scores range from 0 (worst performance) to 3 (best performance) and are based on timed gait speed over 4 metres; standing balance with feet positioned parallel, semi-tandem, and tandem; and standing up from a chair 5 times. The SPPB has high levels of validity, reliability, and responsiveness in measuring mobility in older people living in the community, is increasingly used in trials involving older adults [19], and can predict falls risk, disability, and death [20]. The 12-point version of the SPPB is most commonly used, and 0.5- to 1-point changes have been suggested to be clinically meaningful. We used the continuous version as it has been suggested as more likely to be able to detect change [18].

Physical activity was assessed over a 7-day period at the end of the 6-month intervention period using the activPAL activity monitor (PAL Technologies, UK) [21]. The measure of physical activity was 'upright time', defined as the average proportion of the day spent standing and stepping, measured in 10-second minimum periods. Upright time was chosen as our primary physical activity measure, rather than steps per day, as not all trial participants were expected to be able to walk independently, and we sought to use a measure that could be used at all study time points.

**Secondary outcomes.** Secondary outcomes were performance-based measures assessed at 3 weeks and 6 months after randomisation and participant-reported measures assessed at 3 and 12 weeks and 6 months after randomisation. Performance-based measures of mobility included SPPB (continuous) at 3 weeks; SPPB total score (0 to 12 based on categorisation of performance times; higher score indicates better mobility; clinically important difference 0.5 points) [15,20] and subscale scores (0 to 4) [19]; de Morton Mobility Index (0 to 100; higher score indicates better mobility; clinically important difference 7 to 8 points) [22–24]; single leg stance (0 to 10 seconds; greater time indicates better mobility); maximal balance range test

(millimetres; greater distance indicates better mobility) [25]; and step test (number of steps; greater number of steps indicates better mobility) [26]. Performance-based measures of physical activity included proportion of the day spent upright at 3 weeks, average time spent standing and stepping, number of steps per day, and number of sit to stand transitions per day measured using the activPAL [21]. Performance-based measures of cognition included Trail Making Test A, B, and B − A (seconds; quicker time indicates improved cognition) [27,28].

Participant-reported measures included Incidental and Planned Exercise Questionnaire (IPEQ) total score and home exercise and walking activity subscale scores (hours/week) [29]; Modified Computer Self Efficacy Scale (10 to 100; higher score indicates improved device self-efficacy) [30]; Activities-specific Balance Confidence Scale (0 to 100; higher score indicates improved confidence) [31]; WHO Disability Assessment Schedule 2.0 (12 to 60; lower score indicates improved activity performance and participation) [32,33]; Short Form 6 dimensions questionnaire subscale scores and health utility score (0 to 1; higher score indicates better quality of life; mean minimal important difference 0.041) [34,35]; and European Quality of Life–5 dimensions subscale scores, visual analogue scale score (0 to 100), and health utility score (−0.68 to 1; higher score indicates better quality of life; minimal important difference 0.074) [35,36]. In addition, falls and health and community service usage were assessed over the 6-month period. Adverse events in the intervention group and deaths in both groups were monitored and documented throughout the trial. Adverse events were defined as an unwanted and usually harmful outcome (e.g., fall, seizure, cardiac event) that may or may not be related to the intervention, but occurred while the participant was undertaking mobility or physical activities using intervention digital devices. Self-reported measures of device usability (System Usability Scale; 0 to 100; score above 70 indicates above average usability) [37,38] and enjoyment (Physical Activity Enjoyment Scale; 18 to 126; higher score indicates more enjoyment) [39] were obtained from the intervention group at 3 and 12 weeks and 6 months after randomisation.

### Data analysis

We estimated that a sample size of 300 participants (150 per group) would provide 90% power to detect a 15% between-group difference in the co-primary outcome measures, allowing for a 20% dropout rate and an alpha of 5%. This sample size was also estimated to be sufficient to detect between-group differences of 10%–15% in most secondary outcomes and was considered by the authors to be of meaningful size on the basis of our collective clinical experience with the measures.

A statistical analysis plan was approved by the study statistician (SH) and chief investigator (CS) before data analysis, and no changes were made after this time (see S3 Text). Analysis was conducted by 2 investigators (CS, LH) blinded to group allocation for the co-primary outcomes using dummy codes for group allocation, created by a person external to the trial. The dataset analysed consisted of all randomised participants irrespective of intervention adherence (intention-to-treat). Missing values were not imputed for the primary analyses. Between-group comparisons for continuously scored outcomes were made using linear models with baseline scores entered as covariates. The distribution of continuous variables was evaluated to inform whether change scores were used for analysis. Fall rates between groups were compared using negative binomial regression. Two pre-specified sensitivity analyses were conducted for the co-primary outcomes; (i) not adjusting for baseline scores and (ii) adjusting for stratification variables. *p*-Values were not adjusted for multiplicity as we pre-specified that a significant effect must be observed on both primary outcomes to declare the intervention effective.

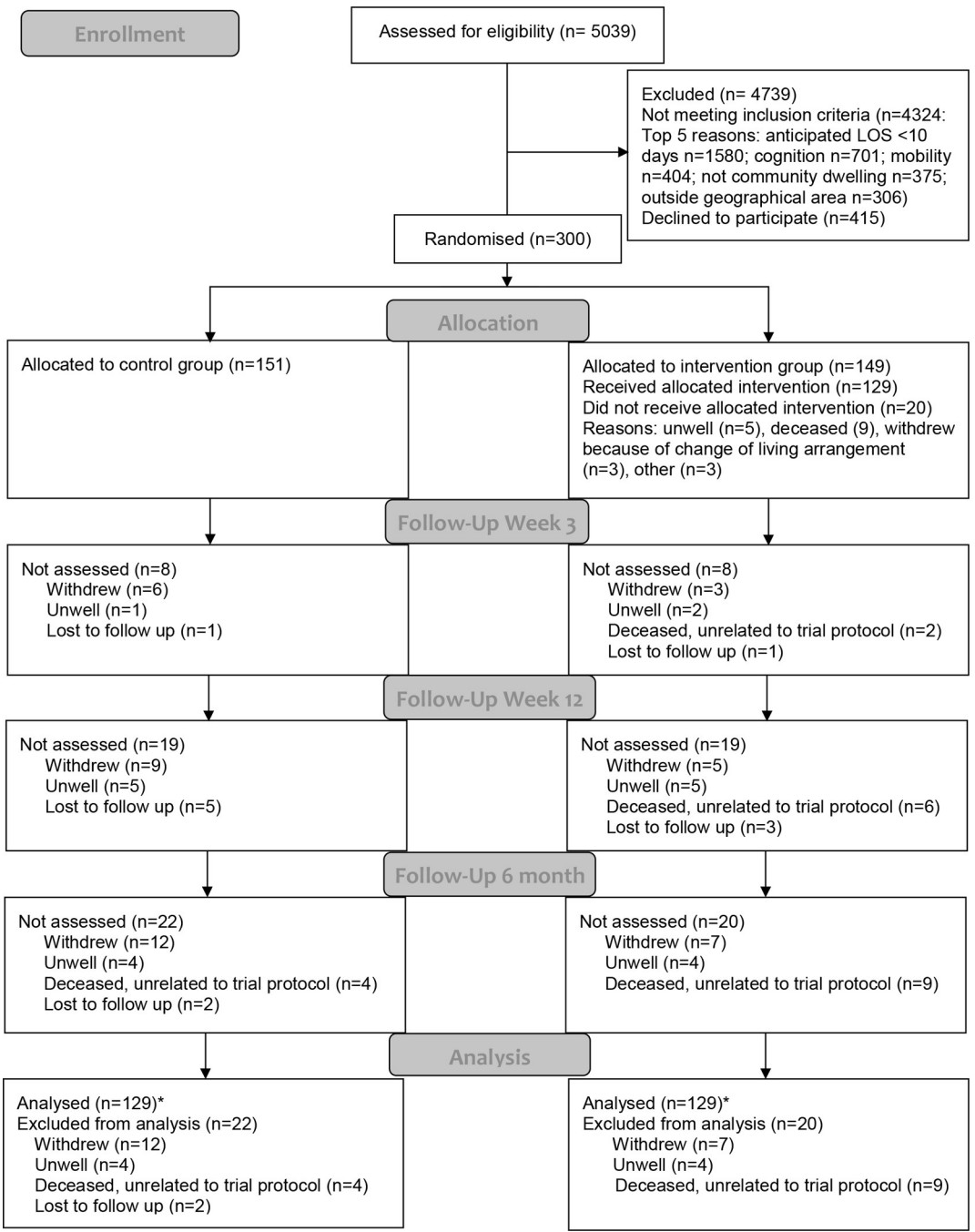

**Fig 1. CONSORT flow diagram.** *Number included in intention-to-treat analysis. LOS, length of stay.

We undertook 6 pre-specified subgroup analyses based on neurological versus non-neuro-logical health conditions limiting mobility, sex, age, baseline mobility (SPPB total score), device use before hospitalisation, and state (New South Wales versus South Australia). The main analysis for each subgroup analysis was an interaction test in the regression models to determine whether the effect of treatment differed significantly across categories for that variable. Analyses were performed using Stata software, version 14 (StataCorp).

**Table 2. Characteristics of participants at baseline.**

| Characteristic | Intervention group *n* = 149 | Control group *n* = 151 |
|---|---|---|
| **Demographics** | | |
| Age (years), mean (SD); range | 70 (18); 18–101 | 73 (15); 21–95 |
| <50, *n* (%) | 21 (14) | 15 (10) |
| 50–69, *n* (%) | 44 (30) | 38 (25) |
| 70–89, *n* (%) | 73 (49) | 85 (57) |
| 90+, *n* (%) | 11 (7) | 13 (8) |
| Sex female, *n* (%) | 72 (48) | 77 (51) |
| Prior living arrangement, *n* (%) | | |
| Alone | 58 (39) | 46 (31) |
| Family | 89 (60) | 102 (68) |
| Non-relative | 2 (1) | 3 (1) |
| Marital status, *n* (%) | | |
| Currently married/cohabitating | 70 (47) | 77 (51) |
| Divorced/separated | 23 (16) | 14 (9) |
| Widowed | 39 (26) | 43 (29) |
| Never married | 17 (11) | 17 (11) |
| Years of education, mean (SD); range | 12 (3); 5–20 | 12 (4); 4–32 |
| 0–12 years, *n* (%) | 85 (57) | 91 (60) |
| 13–16 years, *n* (%) | 39 (26) | 36 (24) |
| >16 years, *n* (%) | 15 (10) | 17 (11) |
| Unknown, *n* (%) | 10 (7) | 7 (5) |
| Current work status, *n* (%) | | |
| Retired | 91 (62) | 95 (63) |
| Paid work | 27 (18) | 22 (15) |
| Homemaker | 6 (4) | 14 (9) |
| Unemployed | 14 (9) | 10 (7) |
| Student | 5 (3) | 2 (1) |
| Volunteer/other | 6 (4) | 8 (5) |
| English primary language at home, *n* (%) | 129 (87) | 129 (85) |
| **Health** | | |
| Neurological condition causing activity limitation, *n* (%) | 80 (54) | 82 (54) |
| Primary diagnosis grouping, *n* (%) | | |
| Neurological | 72 (48) | 77 (51) |
| Cardiopulmonary | 16 (11) | 9 (6) |
| Musculoskeletal | 41 (28) | 48 (32) |
| Restorative care/other | 20 (13) | 17 (11) |
| MMSE score (0–30), mean (SD); range | 27 (3); 15–30 | 27 (3); 17–30 |
| Number of co-morbidities (0–26), mean (SD); range | 5 (3); 0–14 | 5 (3); 0–11 |
| Number of medications at entry to study, mean (SD); range | 8 (3); 1–19 | 9 (3); 2–17 |
| **Function** | | |
| Walking status prior to hospitalisation, *n* (%) | | |
| Did not walk | 0 (0) | 1 (1) |
| Indoor walker only | 17 (11) | 20 (13) |
| Community walker | 132 (89) | 130 (86) |
| **Devices** | | |
| Devices used in month prior to hospitalisation, *n* (%) | | |

*(Continued)*

**Table 2.** (Continued)

| Characteristic | Intervention group $n = 149$ | Control group $n = 151$ |
| --- | --- | --- |
| Computer | 60 (40) | 63 (42) |
| Tablet | 44 (30) | 35 (23) |
| Smartphone | 55 (37) | 52 (34) |
| Gaming console | 6 (4) | 1 (1) |
| Activity monitor | 7 (5) | 2 (1) |

MMSE, Mini-Mental State Examination.

## IRB approval

Two human research ethics committees (HRECs) approved the trial (Southern Adelaide Clinical HREC and South Western Sydney Local Health District HREC). Six minor protocol amendments were approved by the ethics committees, 4 prior to the trial commencing (see S2 Text). We prospectively registered the trial with the Australian New Zealand Clinical Trials Registry (ACTRN12614000936628).

## Results

Between September 2014 and November 2016, 5,039 patients were screened, 715 patients were assessed as eligible, and 300 patients provided written informed consent and were randomised: 149 to the intervention group and 151 to the control group (Fig 1). Six-month assessments were completed by 258 participants (control group: 129/151, 85%; intervention group: 129/149, 87%). For the co-primary outcomes, there was an 85% (254/300) follow-up rate for mobility (data unavailable for 4 additional participants who refused to complete 1 or more test components) and an 80% (239/300) follow-up rate for upright time (data missing or excluded for 19 additional participants due to <4 days wear time for activPAL device, $n = 3$; refusal/unable to wear device, $n = 5$; device initialisation/fault, $n = 3$; device lost, $n = 4$; missing data, $n = 4$).

Baseline characteristics are presented in Tables 2 and S1. On average, participants spent 13 days in the ward before randomisation (SD 16; median 8). Participants had a mean age of 74 (SD 14) years, 50% were female, and 54% had neurological health conditions causing activity limitation. At baseline, participants had significant mobility limitation (mean [SD] SPPB total score 4.2 [2.6]) and spent little time standing or stepping (mean [SD] upright time 112 [90] minutes) (Table 3). Prior to hospital admission, 87% of participants could walk independently in the community and all but 1 could walk indoors. Thirty-nine percent of participants reported never using a computer, tablet, smartphone, gaming device, or activity monitor in the month prior to hospitalisation.

### Intervention fidelity, acceptability, enjoyment, and adherence

Over the 6-month trial period, participants spent on average 19 days (SD 20; median 12) in an inpatient setting and 161 days (SD 18) in a post-hospital setting, typically at home. The total cost of the intervention (staff training, equipment, intervention preparation, and delivery) per participant was AU$1,892 (S2 Table). Intervention data are presented in Table 4. Intervention participants rated the usability of prescribed devices above average, and enjoyment as high at all time points (Table 3).

Participants in both groups received a similar number of usual care physiotherapy sessions in the post-hospital setting (mean [SD]: intervention group, 10 [15]; control group, 10 [13]

**Table 3. Primary and secondary outcome measures at baseline, 3 weeks, 12 weeks, and 6 months.**

| Outcome | Mean (SD), n | | | | | | | |
| --- | --- | --- | --- | --- | --- | --- | --- | --- |
| | Intervention group | | | | Control group | | | |
| | Baseline | 3 weeks | 12 weeks | 6 months | Baseline | 3 weeks | 12 weeks | 6 months |
| *Performance-based outcomes* | | | | | | | | |
| **Physical activity (activPAL)** | | | | | | | | |
| Proportion of the day spent upright (%) | 8.0 (6.7) | 14.5 (8.4) | | 18.2 (9.8) | 7.5 (5.7) | 14.2 (8.6) | | 18.4 (10.2) |
| | n = 146 | n = 135 | | n = 121 | n = 151 | n = 141 | | n = 124 |
| Time spent upright (minutes/day) | 115 (96) | 208 (122) | | 262 (142) | 109 (83) | 204 (124) | | 265 (147) |
| | n = 146 | n = 135 | | n = 121 | n = 151 | n = 141 | | n = 124 |
| Time spent standing (minutes/day) | 97 (91) | 164 (105) | | 201 (121) | 87 (74) | 161 (104) | | 209 (122) |
| | n = 146 | n = 135 | | n = 121 | n = 151 | n = 141 | | n = 124 |
| Time spent stepping (minutes/day) | 19 (17) | 44 (30) | | 61 (40) | 21 (23) | 43 (33) | | 56 (38) |
| | n = 146 | n = 135 | | n = 121 | n = 151 | n = 141 | | n = 124 |
| Number of steps per day | 1,107 (1,101) | 2,892 (2,144) | | 4,395 (3,129) | 1,315 (1,754) | 2,865 (2,590) | | 3,858 (2,951) |
| | n = 146 | n = 135 | | n = 121 | n = 151 | n = 141 | | n = 124 |
| Number of sit to stand transitions per day | 36 (18) | 42 (14) | | 43 (16) | 38 (24) | 43 (19) | | 41 (15) |
| | n = 146 | n = 135 | | n = 121 | n = 151 | n = 141 | | n = 124 |
| **Mobility** | | | | | | | | |
| Short Physical Performance Battery | | | | | | | | |
| Continuous (0–3) | 1.5 (0.7) | 2.1 (0.6) | | 2.3 (0.6) | 1.5 (0.8) | 1.8 (0.8) | | 2.1 (0.8) |
| | n = 149 | n = 139 | | n = 126 | n = 149 | n = 141 | | n = 129 |
| Total score (0–12) | 4.3 (2.6) | 6.7 (2.9) | | 7.9 (3.1) | 4.2 (2.6) | 5.8 (3.3) | | 7.0 (3.4) |
| | n = 149 | n = 141 | | n = 128 | n = 151 | n = 143 | | n = 129 |
| Balance subscale (0–4) | 2.2 (1.5) | 3.0 (1.3) | | 3.3 (1.1) | 2.0 (1.4) | 2.6 (1.4) | | 3.0 (1.3) |
| | n = 149 | n = 141 | | n = 128 | n = 151 | n = 143 | | n = 129 |
| Gait speed subscale (0–4) | 1.6 (1.1) | 2.5 (1.2) | | 2.9 (1.1) | 1.6 (1.2) | 2.2 (1.3) | | 2.7 (1.3) |
| | n = 149 | n = 141 | | n = 128 | n = 151 | n = 143 | | n = 129 |
| Chair stand subscale (0–4) | 0.5 (0.8) | 1.2 (1.3) | | 1.7 (1.5) | 0.6 (1.0) | 1.0 (1.2) | | 1.4 (1.3) |
| | n = 149 | n = 141 | | n = 128 | n = 151 | n = 143 | | n = 129 |
| de Morton Mobility Index (0–100) | 45.3 (12.2) | 58.9 (15.3) | | 67.4 (18.3) | 44.3 (13.4) | 54.2 (19.2) | | 64.4 (19.6) |
| | n = 149 | n = 141 | | n = 128 | n = 151 | n = 143 | | n = 128 |
| Single leg stance (0–10 seconds) | 1.9 (3.3) | 3.7 (4.1) | | 5.4 (4.3) | 2.1 (3.3) | 2.9 (3.8) | | 4.2 (4.2) |
| | n = 149 | n = 141 | | n = 127 | n = 151 | n = 143 | | n = 129 |
| Maximal balance range test (millimetres) | 101.8 (63.0) | 129.2 (64.5) | | 143.4 (76.8) | 97.4 (61.7) | 110.8 (69.6) | | 125.7 (67.1) |
| | n = 149 | n = 141 | | n = 128 | n = 151 | n = 143 | | n = 129 |
| Step test (steps, average of both legs) | 4.2 (4.9) | 7.7 (5.4) | | 10.1 (5.9) | 4.0 (5.0) | 6.0 (5.8) | | 8.2 (6.1) |
| | n = 149 | n = 141 | | n = 128 | n = 151 | n = 143 | | n = 129 |
| **Cognition**[†] | | | | | | | | |
| Trail Making Test A (0–120 seconds) | 59.3 (29.5) | 45.6 (21.7) | | 43.3 (22.5) | 62.4 (31.7) | 51.3 (27.7) | | 45.1 (22.9) |
| | n = 149 | n = 141 | | n = 128 | n = 151 | n = 142 | | n = 127 |
| Trail Making Test B (0–300 seconds) | 165.6 (91.8) | 121.6 (73.1) | | 107.7 (69.4) | 173.7 (90.8) | 127.3 (78.7) | | 110.4 (62.1) |
| | n = 149 | n = 141 | | n = 128 | n = 151 | n = 142 | | n = 126 |
| Trail Making Test B minus A (seconds) | 106.3 (74.5) | 75.9 (58.8) | | 64.5 (51.8) | 111.3 (71.0) | 76.1 (57.0) | | 65.9 (48.4) |
| | n = 149 | n = 141 | | n = 128 | n = 151 | n = 142 | | n = 126 |
| *Participant-reported outcome measures* | | | | | | | | |
| Incidental and Planned Exercise Questionnaire (hours/week) | | | | | | | | |

*(Continued)*

**Table 3.** (*Continued*)

| Outcome | Mean (SD), *n* | | | | | | | |
|---|---|---|---|---|---|---|---|---|
| | **Intervention group** | | | | **Control group** | | | |
| | **Baseline** | **3 weeks** | **12 weeks** | **6 months** | **Baseline** | **3 weeks** | **12 weeks** | **6 months** |
| Total score | | 20.9 (14.7) | 23.0 (16.3) | 27.0 (15.3) | | 19.2 (12.8) | 21.9 (18.1) | 24.6 (16.1) |
| | | *n* = 140 | *n* = 128 | *n* = 128 | | *n* = 143 | *n* = 127 | *n* = 129 |
| Home exercise subscale | | 1.6 (2.9) | 1.5 (2.6) | 1.8 (3.2) | | 1.9 (3.3) | 1.5 (2.9) | 1.3 (2.4) |
| | | *n* = 140 | *n* = 128 | *n* = 128 | | *n* = 143 | *n* = 127 | *n* = 129 |
| Walking activity subscale | | 2.7 (3.5) | 3.3 (4.0) | 4.8 (5.8) | | 1.7 (2.4) | 2.3 (4.5) | 2.7 (3.6) |
| | | *n* = 140 | *n* = 128 | *n* = 128 | | *n* = 143 | *n* = 127 | *n* = 129 |
| Modified Computer Self Efficacy Scale (10–100) | 65.0 (22.1) | 67.8 (26.8) | 66.0 (27.8) | 75.1 (24.3) | 62.3 (23.6) | 70.3 (24.9) | 65.4 (26.4) | 70.8 (26.1) |
| | *n* = 149 | *n* = 141 | *n* = 130 | *n* = 129 | *n* = 151 | *n* = 143 | *n* = 132 | *n* = 127 |
| Activities-specific Balance Confidence Scale (0–100) | 39.6 (26.6) | 51.7 (26.1) | 57.3 (26.0) | 66.5 (23.6) | 36.3 (26.5) | 49.7 (27.2) | 55.3 (30.2) | 62.4 (26.8) |
| | *n* = 148 | *n* = 141 | *n* = 129 | *n* = 129 | *n* = 151 | *n* = 143 | *n* = 132 | *n* = 128 |
| WHO Disability Assessment Schedule 2.0 (raw score 12–60)[†] | | 27.8 (7.8) | 25.6 (8.5) | 21.8 (7.4) | | 29.2 (8.2) | 26.5 (9.7) | 23.1 (8.6) |
| | | *n* = 141 | *n* = 131 | *n* = 129 | | *n* = 143 | *n* = 132 | *n* = 128 |
| Short Form 6 dimensions questionnaire | | | | | | | | |
| Physical function domain (1–6) | 4.4 (1.1) | 4.0 (0.9) | 3.7 (1.0) | 3.6 (1.1) | 4.5 (1.1) | 4.1 (0.9) | 3.8 (1.2) | 3.6 (1.2) |
| | *n* = 149 | *n* = 141 | *n* = 130 | *n* = 129 | *n* = 150 | *n* = 143 | *n* = 132 | *n* = 129 |
| Role limitation domain (1–4) | 3.1 (1.1) | 3.3 (1.0) | 3.2 (1.1) | 2.8 (1.1) | 3.3 (1.0) | 3.1 (1.0) | 3.1 (1.1) | 2.9 (1.2) |
| | *n* = 149 | *n* = 141 | *n* = 130 | *n* = 129 | *n* = 150 | *n* = 143 | *n* = 132 | *n* = 129 |
| Social functioning domain (1–5) | 3.2 (1.6) | 3.2 (1.4) | 2.5 (1.3) | 2.1 (1.3) | 3.3 (1.6) | 3.1 (1.6) | 2.6 (1.5) | 2.3 (1.5) |
| | *n* = 149 | *n* = 141 | *n* = 130 | *n* = 129 | *n* = 150 | *n* = 142 | *n* = 132 | *n* = 128 |
| Pain domain (1–6) | 3.4 (1.8) | 3.3 (1.7) | 3.2 (1.6) | 2.8 (1.4) | 3.9 (1.6) | 3.2 (1.6) | 3.3 (1.5) | 3.0 (1.5) |
| | *n* = 149 | *n* = 141 | *n* = 130 | *n* = 129 | *n* = 150 | *n* = 143 | *n* = 132 | *n* = 129 |
| Mental health domain (1–5) | 2.6 (1.2) | 2.4 (1.2) | 2.3 (1.2) | 2.2 (1.2) | 2.6 (1.1) | 2.6 (1.2) | 2.5 (1.2) | 2.3 (1.3) |
| | *n* = 149 | *n* = 141 | *n* = 130 | *n* = 129 | *n* = 150 | *n* = 143 | *n* = 132 | *n* = 129 |
| Vitality domain (1–5) | 3.6 (1.3) | 3.4 (1.1) | 3.3 (1.1) | 3.1 (1.0) | 3.8 (1.2) | 3.6 (1.2) | 3.5 (1.1) | 3.3 (1.1) |
| | *n* = 149 | *n* = 141 | *n* = 129 | *n* = 129 | *n* = 150 | *n* = 143 | *n* = 132 | *n* = 129 |
| Health utility (0–1) | 0.28 (0.26) | 0.32 (0.25) | 0.38 (0.24) | 0.45 (0.25) | 0.22 (0.24) | 0.30 (0.26) | 0.35 (0.29) | 0.42 (0.30) |
| | *n* = 149 | *n* = 141 | *n* = 129 | *n* = 129 | *n* = 150 | *n* = 142 | *n* = 132 | *n* = 128 |
| EuroQOL-5L | | | | | | | | |
| Mobility domain (1–5) | 3.0 (1.0) | 2.3 (1.0) | 2.3 (1.0) | 2.0 (1.0) | 2.9 (1.1) | 2.4 (1.1) | 2.5 (1.1) | 2.2 (1.0) |
| | *n* = 149 | *n* = 141 | *n* = 130 | *n* = 129 | *n* = 151 | *n* = 143 | *n* = 132 | *n* = 129 |
| Selfcare domain (1–5) | 2.4 (1.2) | 1.8 (1.0) | 1.7 (0.9) | 1.5 (0.9) | 2.5 (1.1) | 2.0 (1.0) | 1.8 (1.1) | 1.7 (1.1) |
| | *n* = 149 | *n* = 141 | *n* = 130 | *n* = 129 | *n* = 151 | *n* = 143 | *n* = 132 | *n* = 129 |
| Usual activities domain (1–5) | 3.2 (1.4) | 2.7 (1.2) | 2.4 (1.2) | 1.9 (0.9) | 3.5 (1.3) | 2.8 (1.3) | 2.6 (1.3) | 2.1 (1.2) |
| | *n* = 149 | *n* = 140 | *n* = 130 | *n* = 129 | *n* = 151 | *n* = 143 | *n* = 132 | *n* = 129 |
| Pain or discomfort domain (1–5) | 2.4 (1.1) | 2.0 (1.0) | 2.2 (1.1) | 2.0 (0.9) | 2.6 (1.1) | 2.2 (1.1) | 2.3 (1.0) | 2.1 (1.0) |
| | *n* = 149 | *n* = 141 | *n* = 129 | *n* = 129 | *n* = 151 | *n* = 143 | *n* = 132 | *n* = 129 |
| Anxiety or depression domain (1–5) | 1.8 (1.0) | 1.6 (0.9) | 1.7 (0.9) | 1.6 (0.9) | 1.8 (0.9) | 1.7 (0.9) | 1.8 (1.0) | 1.6 (0.8) |
| | *n* = 149 | *n* = 141 | *n* = 130 | *n* = 129 | *n* = 151 | *n* = 143) | *n* = 132 | *n* = 129 |
| VAS score (0–100) | 54.5 (21.9) | 65.7 (18.3) | 66.9 (20.8) | 71.5 (18.3) | 55.0 (20.7) | 64.3 (22.1) | 67.2 (20.3) | 70.2 (20.7) |
| | *n* = 149 | *n* = 141 | *n* = 130 | *n* = 129 | *n* = 151 | *n* = 143 | *n* = 132 | *n* = 129 |

(*Continued*)

**Table 3.** (Continued)

| Outcome | Mean (SD), *n* | | | | | | | |
| --- | --- | --- | --- | --- | --- | --- | --- | --- |
| | Intervention group | | | | Control group | | | |
| | Baseline | 3 weeks | 12 weeks | 6 months | Baseline | 3 weeks | 12 weeks | 6 months |
| Health utility score (−0.68 to 1) | 0.40 (0.36) | 0.60 (0.27) | 0.58 (0.29) | 0.70 (0.25) | 0.36 (0.29) | 0.54 (0.31) | 0.52 (0.35) | 0.65 (0.29) |
| | *n* = 149 | *n* = 140 | *n* = 129 | *n* = 129 | *n* = 151 | *n* = 143 | *n* = 132 | *n* = 129 |
| System Usability Scale (0–100) | | 72.2 (18.7) | 74.2 (19.8) | 78.0 (17.4) | | | | |
| | | *n* = 134 | *n* = 123 | *n* = 127 | | | | |
| Physical Activity Enjoyment Scale (18–126) | | 95.5 (23.2) | 95.7 (22.0) | 98.3 (20.8) | | | | |
| | | *n* = 133 | *n* = 122 | *n* = 127 | | | | |

[†]A lower score indicates a better performance.

EuroQOL-5L, European Quality of Life–5; VAS, visual analogue scale.

sessions). Few control participants reported using devices for mobility or physical activity (inpatient setting: computer, *n* = 1; tablet, *n* = 2; activity monitor, *n* = 3; post-hospital setting: smartphone, *n* = 1; gaming device, *n* = 2; activity monitor, *n* = 9 participants).

## Effect of intervention

**Co-primary outcomes.** Change in mobility scores were higher in the intervention group compared to the control group from baseline (SPPB [continuous, 0–3] mean [SD]: intervention group, 1.5 [0.7]; control group, 1.5 [0.8]) to 6 months (mean between-group difference 0.2 points, 95% CI 0.1 to 0.3; *p* = 0.006); however, there was no evidence of a difference between groups for upright time at 6 months (mean [SD] proportion of the day spent upright at 6 months: intervention group, 18.2 [9.8]; control group, 18.4 [10.2]; mean between-group difference −0.2, 95% CI −2.7 to 2.3; *p* = 0.87), with similar results in sensitivity analyses (S3 Table) and at week 3 (Table 5).

**Secondary outcomes.** There were between-group differences in favour of the intervention group across most secondary mobility outcomes (Table 5), for change in self-reported time spent walking from 3 weeks to 6 months (IPEQ walking activity subscale, hours/week: 1.8, 95% CI 0.6 to 3.0, *n* = 254; *p* = 0.004), and for change on 1 measure of cognition from baseline to 3 weeks (Trail Making Test A: −5.1 seconds, 95% CI −9.3 to −0.8, *n* = 283; *p* = 0.02). There was no evidence of a difference between groups in the number of steps taken per day from baseline to 6 months (mean between-group difference 646 steps per day, 95% CI −109 to 1,402, *n* = 239; *p* = 0.09) or on any other secondary outcomes (Tables 5 and 6 and S4). Thirteen participants died while in the trial (intervention group: 9; control group: 4) due to causes unrelated to the trial. The same number of participants reported falling 1 or more times in both groups (*n* = 53), and there was no difference between groups in fall rate (S5 Table). No adverse events, defined as incidents that occurred while participating in the intervention, were reported.

Interaction analysis for primary outcomes indicated a greater effect of the intervention on mobility among those with poorer mobility at baseline (Tables 5 and S6). Exploratory analyses for secondary outcomes revealed consistently greater intervention impact in younger participants (Tables 5 and 6 and S7).

**Table 4. Intervention group data.**

| Characteristic | Mean (SD), percent, or n (%) |
|---|---|
| *Inpatient (n = 149)* | |
| **Dose** | |
| Number sessions offered | 11 (16)[#] |
| Number sessions delivered | 7 (10)[#] |
| Duration of sessions, minutes | 41 (11) |
| **Reasons for sessions not delivered** | |
| Day of discharge | 18% |
| Feeling tired/unwell | 16% |
| Refusal | 11% |
| Unknown | 11% |
| Public holiday | 10% |
| **Devices used** | |
| Number of devices | 4 (1) |
| Nintendo Wii | 36 (24%) |
| Xbox Kinect | 39 (26%) |
| Activity monitor (Fitbit, Garmin) | 120 (81%) |
| Smartphone physical activity app | 3 (2%) |
| Fysiogaming | 85 (57%) |
| iPad exercise app | 107 (72%) |
| Humac Balance System | 89 (60%) |
| Stepping Tiles | 46 (31%) |
| **Mobility limitations addressed using devices** | |
| Maintaining standing position | 120 (81%) |
| Stepping while standing | 119 (80%) |
| Standing up from a chair | 114 (77%) |
| Reaching while standing | 67 (45%) |
| Changing directions while walking | 56 (38%) |
| Stair climbing | 25 (17%) |
| Physical activity through the day | 135 (91%) |
| *Community (n = 144)* | |
| **Dose** | |
| Number contacts with physiotherapist | 15 (5) |
| Home visit frequency | 6 (1) |
| Home visit duration, minutes | 46 (13) |
| Phone call frequency | 8 (4) |
| Phone call duration, minutes | 8 (3) |
| Other* frequency | 1 (1) |
| Other* duration, minutes | 6 (20) |
| **Reason for physiotherapist contact** | |
| Health coaching | 68% |
| Quick contact | 20% |
| Device support | 8% |
| Other | 4% |
| **Devices used** | |
| Number of devices | 2 (1) |
| Nintendo Wii | 23 (16%) |
| Xbox Kinect | 24 (17%) |

(*Continued*)

**Table 4.** (Continued)

| Characteristic | Mean (SD), percent, or *n* (%) |
|---|---|
| Activity monitor (Fitbit, Garmin) | 141 (98%) |
| Smartphone physical activity app | 8 (6%) |
| Fysiogaming (home version) | 5 (3%) |
| iPad exercise app | 124 (86%) |
| **Topics covered in health coaching sessions (*n* = 1,419 sessions)** | |
| Objective data from devices | 1128 (80%) |
| Physical activity status | 999 (70%) |
| Mobility status | 994 (70%) |
| Adherence (barriers and facilitators) | 909 (64%) |
| Goal setting and evaluation | 662 (47%) |
| Technical issues and assistance | 537 (38%) |
| Modification of exercise program | 495 (35%) |
| Physical activity/health education | 296 (21%) |
| Fall prevention and education | 225 (16%) |
| Other | 210 (15%) |
| ***6-month physiotherapist-rated level of adherence*** | |
| >75% | 45 (30%) |
| 50–74% | 37 (25%) |
| 25–49% | 30 (25%) |
| 1–24% | 25 (17%) |
| 0% | 7 (5%) |
| Not rated | 5 (3%) |

[#]Median (IQR) values.

[*]Other: email, video conference, SMS, hospital visit.

## Discussion

We conducted a pragmatic, assessor-blinded, parallel-group randomised trial in people with mobility limitations undertaking aged care and neurological rehabilitation recruited from 3 Australian hospitals to investigate whether tailored prescription of affordable digital devices (including virtual reality video games, activity monitors, and handheld computer devices) in addition to usual care could improve mobility and physical activity when compared with people undertaking usual care alone. There was no evidence of effectiveness of the intervention in accordance with our pre-specified definition that both primary outcomes needed to show statistically significant between-group differences. However, significant and clinically relevant improvements in mobility were observed in participants receiving the AMOUNT intervention. The greatest improvements in mobility were seen at 3 weeks during hospital-supervised therapy. Between-group differences were still evident at 6 months despite the lower intensity physiotherapy support in the post-hospital period. All available devices were used, supporting our premise of a multi-device intervention over using a single device as in previous studies. Six of the devices were used across both inpatient and post-hospital care settings, and usability and enjoyment were rated highly. Taken altogether, these findings suggest that digitally enabled rehabilitation, supported by physiotherapists, is feasible and acceptable and can improve mobility outcomes.

The mean between-group difference on our primary mobility measure at 6 months (0.2 points) may be considered of clinical importance. A change of 0.54 on the 12-point version of the SPPB, i.e., 4.5% of the maximum value, has been suggested to be a small meaningful change

**Table 5. Primary and secondary performance-based outcomes.**

| Outcome | Time point or time between assessments | Mean between-group difference (95% CI) in outcome, adjusted for baseline; n | p-Value |
|---|---|---|---|
| **Co-primary outcomes** | | | |
| **Mobility (positive MD favours intervention group)** | | | |
| SPPB (continuous version, 0–3) | 6 mo minus baseline | 0.2 (0.1 to 0.3); 254[&,§] | 0.006 |
| **Physical activity (positive MD favours intervention group)** | | | |
| Proportion of the day spent upright (%) | At 6 mo | −0.2 (−2.7 to 2.3); 239[§] | 0.87 |
| **Secondary outcomes** | | | |
| **Mobility (positive MD favours intervention group)** | | | |
| SPPB | | | |
| Continuous version (0–3) | 3 wk minus baseline | 0.3 (0.1 to 0.4); 279[&] | <0.001 |
| Total score (0–12) | 3 wk minus baseline | 0.9 (0.3 to 1.5); 284[¶,&] | 0.002 |
| | 6 mo minus baseline | 0.9 (0.2 to 1.6); 257[¶,#,*,§] | 0.01 |
| Balance subscale score (0–4)~ | 3 wk minus baseline | 1.9 (1.2 to 3.1); 284 | 0.007 |
| | 6 mo minus baseline | 1.9 (1.1 to 3.1); 257 | 0.02 |
| Gait speed subscale score (0–4)~ | 3 wk minus baseline | 1.5 (1.0 to 2.3); 284 | 0.07 |
| | 6 mo minus baseline | 1.4 (0.9 to 2.3); 257 | 0.13 |
| Chair stand subscale score (0–4)~ | 3 wk minus baseline | 1.9 (1.2 to 3.0); 284 | 0.006 |
| | 6 mo minus baseline | 1.6 (1.0 to 2.5); 257 | 0.04 |
| de Morton Mobility Index (0–100) | 3 wk minus baseline | 4.0 (0.8 to 7.2); 284[&] | 0.02 |
| | 6 mo minus baseline | 2.8 (−1.2 to 6.9); 256[‡] | 0.17 |
| Single leg stance (0–10 seconds) | 3 wk minus baseline | 0.9 (0.1 to 1.8); 284 | 0.03 |
| | 6 mo minus baseline | 1.2 (0.2 to 2.2); 256[§] | 0.02 |
| Maximal balance range test (millimetres) | 3 wk minus baseline | 16.8 (3.2 to 30.4); 284[&] | 0.02 |
| | 6 mo minus baseline | 17.5 (1.6 to 33.4); 257 | 0.03 |
| Step test (steps, average of both legs) | 3 wk minus baseline | 1.7 (0.6 to 2.7); 284[¶,§] | 0.002 |
| | 6 mo minus baseline | 2.0 (0.7 to 3.3); 257 | 0.003 |
| **Physical activity (positive MD favours intervention group)** | | | |
| Proportion of the day spent upright, percent | At 3wk | 0.2 (−1.8 to 2.1); 271 | 0.86 |
| Time spent upright (minutes/day) | At 3wk | 2.4 (−25.3 to 30.2); 271 | 0.86 |
| | At 6 mo | −3.1 (−39.4 to 33.2); 239[§] | 0.87 |
| Time spent standing (minutes/day) | 3 wk minus baseline | 0.7 (−23.1 to 24.5); 271 | 0.96 |
| | 6 mo minus baseline | −9.3 (−39.7 to 21.1); 239[^] | 0.55 |
| Time spent stepping (minutes/day) | 3 wk minus baseline | 3.2 (−3.1 to 9.6); 271 | 0.32 |
| | 6 mo minus baseline | 6.4 (−3.3 to 16.2); 239[#,§] | 0.19 |
| Number of steps per day | 3 wk minus baseline | 238 (−223 to 699); 271[^] | 0.31 |
| | 6 mo minus baseline | 646 (−109 to 1,402); 239[#,§] | 0.09 |
| Number of sit to stand transitions per day | 3 wk minus baseline | 0 (−4 to 3); 271 | 0.88 |
| | 6 mo minus baseline | 2 (−2 to 6); 239[§] | 0.31 |
| **Cognition (negative MD favours intervention group)** | | | |
| Trail Making Test A (seconds) | 3 wk minus baseline | −5.1 (−9.3 to −0.8); 283[‡,^] | 0.02 |
| | 6 mo minus baseline | −1.3 (−6.6 to 4.0); 255[‡] | 0.64 |
| Trail Making Test B (seconds) | 3 wk minus baseline | 0.4 (−12.7 to 13.5); 283[‡] | 0.95 |
| | 6 mo minus baseline | 4.0 (−10.2 to 18.3); 254 | 0.58 |

(Continued)

**Table 5.** (Continued)

| Outcome | Time point or time between assessments | Mean between-group difference (95% CI) in outcome, adjusted for baseline; *n* | *p*-Value |
|---|---|---|---|
| Trail Making Test B − A (seconds) | 3 wk minus baseline | 1.7 (−8.7 to 12.0); 283[‡] | 0.75 |
| | 6 mo minus baseline | 0.1 (−10.3 to 10.5); 254[¶] | 0.99 |

Unless otherwise indicated, analyses were conducted with linear regression models with baseline scores entered as covariates. Due to skewed distributions, the change score between time points was used for all outcomes except proportion of the day spent upright. Confidence intervals have not been adjusted for multiplicity, so inferences drawn from the intervals may not be reproducible. Between-group differences are presented as odds ratios. Footnotes indicate significant interactions (*p* ≤ 0.05) for the following pre-specified variables at the given time points:

[#]age as a continuous variable;

[*]age dichotomised at the median (76 years);

[&]baseline mobility as a continuous variable (SPPB total score);

[^]prior device use;

[§]state (New South Wales versus South Australia);

[¶]health condition (neurological versus non-neurological);

[‡]sex.

[~]Analyses conducted with ordered logistic regression for final scores, with baseline scores as a covariate.

MD, mean difference; SPPB, Short Physical Performance Battery.

[20]. The between-group difference at 6 months in the present study represents 6.7% of the maximum value for the 3-point version; therefore, it may represent meaningful change.

In the inpatient setting, participants received on average 41 minutes daily of additional rehabilitation using devices (Table 4). Approximately 60% of participants used the rehabilitation video games (Fysiogaming and the Humac Balance System), which enable the greatest customisation of task-specific mobility training. Our findings of improved mobility are consistent with previous systematic reviews demonstrating improved activity when a greater amount of task-specific practice is provided [4,5]. In contrast, our findings of improved mobility are different than those of the Cochrane systematic review of virtual reality interventions in people after stroke for the effect of additional virtual reality intervention on global motor function [11]. This is likely due to our multi-device intervention and detailed intervention protocol, enabling additional task-specific practice of a range of mobility tasks, compared to the lower limb trials in the review using 1 device, typically targeting balance. The range of health conditions and inclusion of younger participants in our trial may also explain the different findings; however, participants with neurological health conditions and participants with worse mobility at baseline had the greatest improvements in mobility (SPPB total score), particularly at 3 weeks. It is difficult to tease out the contributing role of both amount and type of practice; however, our results suggest that attention to quality and quantity of rehabilitation practice is important.

Although the physical capacity of participants in the intervention group to move around improved, this did not translate to increased time spent upright. Yet there was an indication of more steps taken by intervention participants (*p* = 0.09; particularly younger participants, <76 years, *p* = 0.05), greater self-reported walking, and more time spent stepping and less time spent standing compared to control participants. This finding matches the way the intervention was delivered, with a focus on increasing the number of steps per day using a Fitbit tracker, rather than on standing activities. Further exploration of trial activPAL data is underway to better understand our findings and to help determine how best to prescribe physical activity in this population.

The success of the intervention in improving mobility is likely due to the personalisation of the intervention, which targeted each person's mobility limitations. The included devices were

**Table 6. Secondary participant-reported outcomes.**

| Outcome | Time point or time between assessments | Mean between-group difference (95% CI) in outcome, adjusted for baseline; n | p-Value |
|---|---|---|---|
| Incidental and Planned Exercise Questionnaire (positive MD favours intervention group) | | | |
| Total score (h/wk) | 12 wk minus 3 wk | 0.4 (−3.7 to 4.4); 252 | 0.86 |
| | 6 mo minus 3 wk | 1.9 (−1.7 to 5.6); 254 | 0.31 |
| Home exercise subscale score (h/wk) | 12 wk minus 3 wk | 0.1 (−0.6 to 0.8); 252 | 0.79 |
| | 6 mo minus 3 wk | 0.7 (−0.0 to 1.3); 254[#] | 0.05 |
| Walking activity subscale score (h/wk) | 12 wk minus 3 wk | 0.7 (−0.3 to 1.6); 252[&] | 0.19 |
| | 6 mo minus 3 wk | 1.8 (0.6 to 3.0); 254 | 0.004 |
| Modified Computer Self Efficacy Scale (10–100) (positive MD favours intervention group) | 3 wk minus baseline | −4.8 (−9.7 to 0.1); 284 | 0.06 |
| | 12 wk minus baseline | −1.1 (−6.8 to 4.5); 262 | 0.70 |
| | 6 mo minus baseline | 2.2 (−3.3 to 7.7); 256[‡] | 0.43 |
| Activities-specific Balance Confidence Scale (0–100) (positive MD favours intervention group) | 3 wk minus baseline | 0.6 (−4.7 to 5.8); 283 | 0.83 |
| | 12 wk minus baseline | 1.2 (−5.1 to 7.5); 260[§] | 0.71 |
| | 6 mo minus baseline | 4.0 (−1.7 to 9.8); 256 | 0.17 |
| WHO Disability Assessment Schedule 2.0 (raw score 12–60) (negative MD favours intervention group) | 12 wk minus 3 wk | −0.1 (−2.2 to 1.9); 261[#,*,^,§] | 0.89 |
| | 6 mo minus 3 wk | −0.7 (−2.5 to 1.1); 255[‡,§] | 0.46 |
| Short Form 6 dimensions questionnaire (health utility score 0–1) (positive MD favours intervention group) | 3 wk | 0.00 (−0.06 to 0.05); 282 | 0.99 |
| | 12 wk | 0.01 (−0.05 to 0.08); 260[§] | 0.67 |
| | 6 mo | 0.01 (−0.06 to 0.07); 256[‡,§] | 0.82 |
| European Quality of Life–5 dimensions (health utility score −0.68 to 1) (positive MD favours intervention group) | 3 wk minus baseline | 0.04 (−0.02 to 0.10); 283 | 0.15 |
| | 12 wk minus baseline | 0.05 (−0.02 to 0.13); 261[#,*] | 0.16 |
| | 6 mo minus baseline | 0.05 (−0.02 to 0.11); 258[‡] | 0.14 |

This analysis was conducted using linear regression models with baseline scores entered as covariates. Due to skewed distributions, the change score between time points was used for all outcomes except the Short Form 6 dimensions questionnaire. Confidence intervals have not been adjusted for multiplicity, so inferences drawn from the intervals may not be reproducible.

Footnotes indicate significant interactions ($p \leq 0.05$) for the following pre-specified variables at the given time points: [#]age as a continuous variable;

[*]age dichotomised at the median (76 years);

[&]baseline mobility as a continuous variable (SPPB total score);

[^]prior device use;

[§]state (New South Wales versus South Australia);

[¶]health condition (neurological versus non-neurological);

[‡]sex.

MD, mean difference; SPPB, Short Physical Performance Battery.

piloted previously [13], tested by consumer and clinician investigators, and prescribed according to a detailed protocol developed by the investigator team using motor learning principles [40]. The right level of challenge, variety, enjoyment, and support to use the devices appears key to successful participant engagement [14,41].

Study limitations include 15%–19% loss to follow-up at 6 months on co-primary outcomes, as anticipated in this age group of hospitalised patients. Multiplicity is also a consideration due to the number of outcomes measured. Additionally, there was no statistically significant difference in the important participant-reported outcome of health-related quality of life; however, the measures of this outcome were in the direction favouring the intervention group, which may reflect low statistical power to demonstrate significance for this outcome. There was greater time spent with therapists in the intervention group, which could account for the difference between groups. However, as this was a pragmatic trial, we consider our choice of usual care and an enhanced program to be the correct comparison, and our trial found

additional benefits of the enhanced program. Contamination was of concern prior to commencing the study; however, only a small number of control participants reported using devices for mobility or physical activity. Although the range of devices was a strength, accurate documentation of dosage was difficult because of differences in the types of output data (e.g., game time, repetitions), particularly at home. Development and testing of efficient solutions such as clinical dashboards that enable data from diverse sources to be integrated into a common platform [42] may facilitate tailored use and monitoring of multiple devices in rehabilitation.

Further research should investigate whether future models of rehabilitation care can incorporate digital devices to enhance inpatient and post-hospital rehabilitation with a higher dose of practice whilst conserving quality. Hybrid type II effectiveness–implementation study designs [43] could be used to simultaneously test the effectiveness of the clinical intervention (digitally enabled rehabilitation) on patient outcomes and the effectiveness of implementation strategies (e.g., education and training) to support clinicians to include digital devices into practice.

In summary, we observed improved mobility in participants with a wide range of health conditions in a digitally enabled rehabilitation environment, but no between-group differences in upright time. To enhance generalisability, we focussed on devices likely to be affordable for most rehabilitation services, with elements that could transfer into the community when the patient is discharged. Nevertheless, this was a complex intervention, with specialised equipment and expert staff, so further analyses including economic analysis will be important in understanding its acceptability to purchasers and providers of healthcare.

## Supporting information

**S1 CONSORT Checklist.**
(DOCX)

**S1 Table. Participant primary diagnosis at rehabilitation admission.**
(DOCX)

**S2 Table. Costs for digitally enabled rehabilitation intervention.**
(DOCX)

**S3 Table. Sensitivity analyses for primary outcomes.**
(DOCX)

**S4 Table. Primary and secondary outcomes (additional analysis).**
(DOCX)

**S5 Table. Fall outcomes at 26 weeks.**
(DOCX)

**S6 Table. Interaction *p*-values for co-primary outcomes and mean between-group difference (95% CI) for significant ($p \leq 0.05$) interaction terms.**
(DOCX)

**S7 Table. Interaction *p*-values for secondary outcomes and mean-between group difference (95% CI) for significant interaction terms.**
(DOCX)

**S1 Text. Intervention protocol.**
(DOCX)

**S2 Text. AMOUNT rehabilitation trial: Study protocol and intervention protocol amendments.**
(DOC)

**S3 Text. Statistical analysis plan.**
(DOCX)

## Acknowledgments

The authors are grateful to Mr. Ross Pearson for consumer advice and testing different devices. We are also grateful to the study participants, hospital staff, study staff, and students, particularly Ashley Rabie, Elizabeth Lynch, Catherine Kirkham, Areti Dakopoulos, Melani Boyce, Frances Moran, Janine Prestes Vargas, Linda Roylance, Tarcisio Campos Folly, Hannah Kastrappi, Heather Paull, Caroline Hafner, Janette Hall, Anna Miles, Abby Schmidt, and Caitlin Hamilton.

## Author Contributions

**Conceptualization:** Richard I. Lindley, Maria Crotty, Annie McCluskey, Hidde P. van der Ploeg, Stuart T. Smith, Karl Schurr, Catherine Sherrington.

**Data curation:** Leanne Hassett, Maayken van den Berg, Siobhan Wong, Sakina Chagpar, Heather Weber, Marina Pinheiro, Catherine Sherrington.

**Formal analysis:** Leanne Hassett, Richard I. Lindley, Kirsten Howard, Marina Pinheiro, Stephane Heritier, Catherine Sherrington.

**Funding acquisition:** Richard I. Lindley, Maria Crotty, Annie McCluskey, Hidde P. van der Ploeg, Stuart T. Smith, Karl Schurr, Kirsten Howard, Maggie Killington, Bert Bongers, Leanne Togher, Daniel Treacy, Simone Dorsch, Siobhan Wong, Stephane Heritier, Catherine Sherrington.

**Investigation:** Maayken van den Berg, Maggie Killington, Siobhan Wong, Sakina Chagpar, Heather Weber.

**Methodology:** Leanne Hassett, Maayken van den Berg, Richard I. Lindley, Maria Crotty, Annie McCluskey, Hidde P. van der Ploeg, Stuart T. Smith, Karl Schurr, Kirsten Howard, Maree L. Hackett, Maggie Killington, Bert Bongers, Leanne Togher, Daniel Treacy, Simone Dorsch, Katharine Scrivener, Marina Pinheiro, Stephane Heritier, Catherine Sherrington.

**Project administration:** Leanne Hassett, Maayken van den Berg, Catherine Sherrington.

**Resources:** Leanne Hassett, Maayken van den Berg, Stuart T. Smith, Karl Schurr, Bert Bongers, Leanne Togher, Siobhan Wong, Sakina Chagpar, Heather Weber.

**Software:** Leanne Hassett, Maayken van den Berg, Bert Bongers, Catherine Sherrington.

**Supervision:** Leanne Hassett, Maayken van den Berg, Maria Crotty, Catherine Sherrington.

**Validation:** Leanne Hassett, Maayken van den Berg.

**Visualization:** Leanne Hassett, Maayken van den Berg, Catherine Sherrington.

**Writing – original draft:** Leanne Hassett, Maayken van den Berg, Richard I. Lindley, Catherine Sherrington.

**Writing – review & editing:** Leanne Hassett, Maayken van den Berg, Richard I. Lindley, Maria Crotty, Annie McCluskey, Hidde P. van der Ploeg, Stuart T. Smith, Karl Schurr, Kirsten Howard, Maree L. Hackett, Maggie Killington, Bert Bongers, Leanne Togher,

Daniel Treacy, Simone Dorsch, Siobhan Wong, Katharine Scrivener, Sakina Chagpar, Heather Weber, Marina Pinheiro, Stephane Heritier, Catherine Sherrington.

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
