## [Decision Letter · Decision Letter 0]

8 Oct 2019

Dear Dr. Hassett,

Thank you very much for submitting your manuscript "Digitally-enabled rehabilitation to enhance outcomes: The Activity and MObility UsiNg Technology (AMOUNT) randomised controlled trial." (PMEDICINE-D-19-02056) for consideration at PLOS Medicine. 

Your paper was evaluated by a senior editor and discussed among all the editors here. It was also discussed with an academic editor with relevant expertise, and sent to three independent reviewers, including a statistical reviewer. The reviews are appended at the bottom of this email and any accompanying reviewer attachments can be seen via the link below:

[LINK]

In light of these reviews, I am afraid that we will not be able to accept the manuscript for publication in the journal in its current form, but we would like to consider a revised version that addresses the reviewers' and editors' comments. Obviously we cannot make any decision about publication until we have seen the revised manuscript and your response, and we plan to seek re-review by one or more of the reviewers. 

We expect to receive your revised manuscript by Oct 29 2019 11:59PM. Please email us (plosmedicine@plos.org) if you have any questions or concerns.

We look forward to receiving your revised manuscript. 

Sincerely,

Thomas McBride, PhD

Senior Editor 

PLOS Medicine

plosmedicine.org

1- PLOS Medicine requires that the de-identified data underlying the specific results in a published article be made available, without restrictions on access, in a public repository or as Supporting Information at the time of article publication, provided it is legal and ethical to do so. PLOS Medicine does not allow authors to be the primary contact for data access. 

PLOS defines the “minimal data set” to consist of the data set used to reach the conclusions drawn in the manuscript with related metadata and methods, and any additional data required to replicate the reported study findings in their entirety. Authors do not need to submit their entire data set, or the raw data collected during an investigation. Please submit the following data:

The values behind the means, standard deviations and other measures reported;

The values used to build graphs;

The points extracted from images for analysis.

If the deidentified data can not be made freely available, please describe briefly the ethical, legal, or contractual restriction that prevents you from sharing it. Please also include an appropriate contact (web or email address) for inquiries (again, this cannot be a study author).

Please see the policy at 

http://journals.plos.org/plosmedicine/s/data-availability

and FAQs at 

http://journals.plos.org/plosmedicine/s/data-availability#loc-faqs-for-data-policy

2- The title could be a bit more descriptive, specifying the type of rehabilitation.

3- Please combine Methods and Findings sections of Abstract into one section, “Methods and Findings”.

4- Around line 105, please include a new sentence summarizing the secondary outcome findings (from lines 315-328); and a second new sentence summarizing the situation with adverse events, falls and deaths.

5- Please edit your language to be a bit more measureda, e.g. at line 108 "In this study, we observed improved mobility in people with a wide range of health conditions making use of digitally-enabled rehabilitation, whereas time spent upright was unchanged." 

6- In the Methods section, please specify if written or verbal consent was obtained from the study participants, or if the ethics committee waived this requirement.

7- Lines 259-261(“Role of the Funding source”) can be removed.

8- Please briefly describe intervention and trial at beginning of the Discussion.

9- Line 355, please rephrase as "In this study, significant and clinically relevant improvements in mobility were observed in participants receiving the AMOUNT intervention ..." or similar. 

10- At line 391, "we improved" seems too strong, please temper.

11- At line 424, please rephrase to "... we observed improved mobility in participants in a digitally-enabled rehabilitation environment ...". 

Comments from the reviewers:

Reviewer #1: Thank you for the opportunity to review this very interesting RCT evaluating a complex intervention. The authors have conducted a pragmatic trial to evaluate whether digital-enabled rehabilitation can improve mobility and upright time in patients admitted to age and neurological care in hospital settings. Over a six month period, the intervention improved mobility but not upright time. Overall, the analyses were appropriate and followed the pre-specified statistical analyses plan. My comments are specified below:

1) Lines 190-191: Primary outcomes on SPPB - it's not clear why the commonly used 12-point score was not utilised as the primary outcome compared to the 0-3 score. I would think that a 12-point scale would have more variation to allow for movement in the outcome and hence may be able to detect larger changes. The analyses shows this as both the 0-3 score and 12-point score do show significant differences between groups with the different larger in the 12-point score. 

2) Lines 195-198: Primary outcome on activPal - This outcome has been averaged across a 7-day period. I would expect that the daily monitoring would have some large variations relatively which doesn't not necessarily follow a normal distribution. It's also likely reason why it's an average across seven days is not going to capture the variation in the differences between groups. 

3) Outcomes section in methods: The co-primary outcomes although stated in the abstract and findings are described as a mean difference between groups, there is some nuance between the mobility outcome and upright time outcome which could help with better clarity in the methods. The mobility outcome (SPPB) is a mean difference of the relative change from baseline (i.e. difference in difference) between groups whereas the activPAL was only the mean difference between groups at a point prevalence at six months. 

4) Related to point 3) as the tables clearly show that activPAL was also assessed at baseline, did the authors look at the different in the mean difference between groups relative to baseline similar to the method used in analysing SPPB. 

5) The outcome of the upright time has been averaged across a 7-day period for daily time. Was the daily activity time relatively stable as within each individual - as I would expect there is probably some variation which doesn't not necessarily follow a normal distribution? It may be also the reason that the average doesn't capture the nature of the change in this outcome. On suggestion could be look at this as a repeated measure analyses of average per-day between intervention and control groups. 

6) Lines 226-227: It's not clear if falls and health community service usage were also part of adverse events monitored.

7) Lines 246-247: this needs justification using a NBR rather than a Poisson regression. I can venture a guess that a negative binomial model is expected to provide an improved fit to the data and accounted better for over-dispersion (variance is high compared to mean rates) than the Poisson regression model, but authors to clarify. 

8) Consort diagram: I'm going to assume that the n = 129 included in the analysis on the consort was the number analysed for intention to treat - would be helpful to indicate this in the consort diagram with an * footnote

9) Table 2: It would be helpful to see in this baseline characteristics table whether any of the baseline factors were significantly different. There may be some slight variation between intervention and control just on quick view of the table (i.e. proportions for prior living arrangements, primary diagnosis grouping proportions. also intervention group looks like they are experience using devices in month prior to hospital). If some of these factors may be different - it could justify baseline covariate adjustment in the analysis of between group differences as if randomisation didn't completely control for confounding then these factors still may influence the results. 

My overall impression is that this is a well-conducted, interesting, and important piece of work that warrant publication. The authors have stated that the economic analyses will be presented as a separate piece but I would encourage the authors to consider including the cost-effectiveness alongside this RCT as it would help with understanding the value for money of this intervention which would greatly enhance this piece. I see that as a key strength but ultimately up the authors. I hope the authors find the comments useful to help improve their hard work prior to publication. 

Reviewer #2: It was a pleasure to review this well conducted trial evaluating an important clinical intervention, namely the use of digital technology to enhance rehabilitation across the ward and post-discharge settings. The inclusion of the intervention protocol and quality of intervention reporting in the manuscript is commendable. Overall the manuscript is well-written but I have outlined a number of areas that could be addressed to improve precision in the trial reporting. I would encourage revision of the manuscript and I hope my comments are useful to the authors in their endeavours to report this clinically relevant trial.

ABSTRACT

It would be useful to briefly explain what usual care is to provide context

The intervention would benefit from some more specific information as it was more involved than just prescribing devices. 

CONSORT guidance for trial abstracts recommends specifying how participants were allocated to the interventions, further detail would be beneficial

Suggest adding 'outcome' before 'assessor-blinded' to aid clarity.

More information on the eligibility criteria is needed to understand which patient groups this trial would have implications for.

Results

Results for each group are needed with the mean-difference and Cis. 

Suggest alternative wording is used for the results, 'significantly' can be ambiguous - statistically or clinically significant, or both? Suggest 'SPPB scores were higher in the intervention group compared to the control group' (or other wording to that effect).

Rather than 'similar between groups', suggest more precise statistical wording such as 'no evidence of a difference between groups' (or other wording to that effect)

Harms need to be reported, in line with CONSORT.

Suggest removing 'not unexpected in this group of hospitalised patients' as this is more a discussion point.

Conclusion - Were the improvements in SPPB clinically important?

Trial registration number is accurate and consistent with the submitted report

MAIN PAPER

Introduction

114-15: References to support this more general statement relate to stroke, whereas the preceeding sentence suggests this statement is about 'neurological and musculoskeletal health conditions'. References need revisiting or the statement made more precisely.

117-8: what is the hypothesised mechanism by which a digitally enhanced environment enables more practice? Behaviour change? What is the evidence for this mechanism?

126 - sentence tense correction needed 'requires' to 'required'

123-5 - which population was the feasibility assessed in and why this group(s)?

137 - it can be appreciated that the full economic evaluation may be reported later but the cost of delivery would be useful information to present with the main trial results to provide greater context. Not essential but would enhance the report.

146 - what was the trial training? What steps were taken to ensure selection bias was managed (any external audit/screening)?

149-50 - 'clinician assessed capacity for improvement' - how was this assessed? It appears possible that more complex patients could have been systematically excluded? It would be good to report all of the categories for eligibility in the CONSORT rather that the few selected to assess which of these criteria dominated (if any)

155 - what is the definition of a 'high care residential facility' (this is country specific so would be useful context)

165 - The Authors are commended for the reporting of the intervention. Suggest the description of usual care would be useful in the main text in a summary sentence, and a signpost to Table 1, before going onto the intervention.

172 - expand on 'contextual factors'

How did the interventions and protocol ensure sufficient dose and task-specificity? Some information is in the Table 1 and other files but this is important in the main text as it is difficult to interpret the main text independently.

181 - what was the justification for these co-primary outcomes?

192-93 - it is not clear why a different version of the SPPB was selected. The reference cited (Onder) does not seem to explicitly provide strong evidence that supports using this alternative calculation to the otherwise established version - please can the authors expand on this?

227-8 - how were adverse events reported? There is some information in the protocol on definitions but are these self-reported or clinician reported, or both? Differences in levels in clinical contact can influence this event reporting.

236 - why was a 15% between group difference in both measures specified? What were the underlying assumptions on clinically important differences in effects? 

Sample size: It is not clear how the presence of co-primary outcomes was accounted for in the sample size calculation. With co-primary outcomes it is usually assumed that both endpoints need to demonstrate meaningful differences, which reduces power and increases type 2 error. If the sample sizes were calculated separately at 90% power, this would in effectively be 81% power over the two outcomes (0.9 x 0.9 = 0.81). This is likely to be less of an issue as the two endpoints are likely correlated but this section would benefit from further detail, for example, was there any sample size inflation to manage this? I could not find this detail in the protocol, registration, SAP or published protocol.

244 - 'intention to treat' - has this trial used a modified ITT? It looks like it was not a strict ITT as not all participants were included in the final analysis regardless withdrawals or lost to follow-up, please confirm.

247 - please specify of the sensitivity analyses were pre-specified

Were there any protocol amendments? If so, these should be included in the report in line with CONSORT

Much greater detail is needed on the strategies to blind assessors, how were the outcome measures assessed, what safeguards were there?

Results

271/Fig 1 - see earlier comments regarding adding detail to CONSORT on reasons; does the number analysed reflect both primary analyses?

'n=' missing on deceased 

Very low loss to follow-up, excellent - were there any reasons for the withdrawals?

Results - for all percentages, it is informative to provide numerators and denominators

 Table 2 and S1 - balanced baseline characteristics on a whole but note considerable variation in primary condition and age in the trial cohort is evident

302-3 - was there any skew in the number of sessions? Often the case in rehab trials where median, IQR, range is needed - would be reassuring to confirm during review

309 -14 - as per abstract - please see comments on reporting 'significant'

Table 5 - footnote states skew was managed - please add to methods section

325 - secondary outcomes section - with the risk of multiplicity, suggest the focus on evidence and avoiding interpretations, such as 'substantial but not statistically significant'. A focus in the results (effect estimate and measure of uncertainty) would strengthen this section.

349 - an interaction analysis is reported - was this the planned subgroup analyses.

350-2 - there are exploratory analyses presented - the details need to be added to the methods - if post hoc, these need to be justified - could these be reported in another report so that this report focuses on the main trial analyses and outcomes?

Discussion: 

Lines 355 to 358 - as a co-primary outcome was used, should the interpretation start with confirmation that there was no evidence of effectiveness of the intervention? The separate outcomes could then be discussed.

There is a tendency to focus on the significant results/timepoints. Suggest that the statistically significant results should be interpreted with caution as the results section highlights that there were limited differences across most outcomes and timepoints (and the issues with multiplicity - see below).

Limitations: 

-multiplicity is a consideration 

-the great deal of heterogeneity in the devices being tested and in the baseline characteristics of participants included is mentioned but I think the discussion would be enhanced with more discussion around these issues and their implications 

Reviewer #3: Thanks for the opportunity to read your paper. The study is relevant and interesting; the text is concise, methods and results clearly described. I look forward to reading the economic analysis also. 

I have only two items for clarification:

1. Line 441 noted that greater time was spent with therapists in the intervention group than the control group. 

Where are the data are that shows how much additional physiotherapy contact time the intervention group received, in comparison to the control group?

2. Re: Line 355 - the first discussion point regarding …clinically important improvements in …self-reported time spent walking…not accompanied by changes in upright time

Can you comment on the apparent discrepancy between self-reported time spent walking and instrumented (ActivPAL) time spent upright? Also, the comparison between instrumented time stepping and self-reported walking times? 

e.g. Week 3: Instrumented time stepping = 44 min/day; instrumented time upright 208 min/day; IPEQ walking = 23 min/day (2.7 hrs/week)

I would expect self-reported walking and instrumented stepping times to be similar, and upright time (assuming this is standing and walking) to be a lot less.

[LINK]

---

## [Decision Letter · Decision Letter 1]

29 Nov 2019

Dear Dr. Hassett,

Thank you very much for re-submitting your manuscript "Digitally-enabled aged care and neurological rehabilitation to enhance outcomes: The Activity and MObility UsiNg Technology (AMOUNT) randomised controlled trial." (PMEDICINE-D-19-02056R1) for review by PLOS Medicine.

I have discussed the paper with my colleagues and the academic editor and it was also seen again by xxx reviewers. I am pleased to say that provided the remaining editorial and production issues are dealt with we are planning to accept the paper for publication in the journal.

[LINK]

We look forward to receiving the revised manuscript by Dec 06 2019 11:59PM. 

Sincerely,

Louise Gaynor-Brook, MBBS PhD

Associate Editor 

PLOS Medicine

plosmedicine.org

Requests from Editors:

We note the referees comments on inclusion of a cost effectiveness analysis. Having considered and discussed with my colleagues, we feel that inclusion of information on cost evaluation would strengthen the case for publication in PLOS Medicine and we request its addition to the manuscript.

Data Availability: We will require clarification on data availability before publication. If the lead ethics committee do not permit distribution of de-identified data, we will require details for a point of contact (not an author) or a URL where researchers who meet requirements can apply to access the data. We will be unable to proceed with publication until our data requirements are met.

Title: Please revise your title according to PLOS Medicine's style. It should begin with main concept if possible, followed by the study design in the subtitle (ie, after a colon). We suggest ‘Digitally-enabled aged care and neurological rehabilitation to enhance outcomes with Activity and MObility UsiNg Technology (AMOUNT) in Australia: A randomised controlled trial’

Please remove all information within the footer of each page 

Lines 59-86 - Please remove all information presented here

Please be explicit within your Abstract Methods and Findings when recruitment to the RCT took place.

Please add to the Abstract Methods and Findings that the RCT was prospectively registered, providing the trial ID (rather than at the end of Abstract) 

Please add summary demographic information for the participants of the study within your Abstract Methods and Findings 

Line 93 - Please provide details on which cities the hospitals were in 

Line 93 - Please provide details of the age range of adults recruited

Line 100 - Please add (SBBP) after the first mention of Short Physical Performance Battery 

Line 111 - Please add a full stop after ‘...p=0·006)’ to break this up into 2 separate sentences

Line 116 - please give an example or two of what ‘other secondary outcomes’ were 

Line 119 - Please add a sentence to your study limitations that ‘The large number of outcome assessments in our trial increases the risk of a type I error’ or similar. Please also comment on whether provision of the devices could be described as a limitation, relating to possible real-world applicability of the intervention.

Lines 144 & 178 - is there evidence to support that the intervention was enjoyable? If not, please remove.

Line 148 - please omit ‘(0.9 point between group difference on the Short Physical Performance Battery 0 to 12 scale)’ from your Author Summary

Line 175 - please revise to ‘A feasibility trial conducted by our team in people undertaking inpatient aged care and neurological rehabilitation (n=58)’

Line 179 - please omit ‘now’

Please provide dates for recruitment (section beginning line 19) and randomisation (section beginning line 212)

Line 228 – Is the intervention protocol part of the prespecified protocol?

Line 232 – Who paid for the devices that cost $3700? In Table 1 it appears that devices were ‘loaned’; this should also be mentioned in the main text on first mention.

Table 1 & line 286 – For face-to-face sessions and the IPEQ, please provide the questionnaires / interview material (and any others used) as supplementary files. 

Table 1 – Why were the modifications planned? 

Line 312 – Please state modifications to the analysis plan (and why), and refer to the supplementary file in your methods section. 

Line 340 - please clarify whether this was written informed consent 

Table 3 - please provide definitions for all abbreviations used

Tables 5, 6, S2, S3, S5, S6 - When a p value is given, please specify the statistical test used to determine it.

Line 435 - Please revise this paragraph as ‘no evidence of effectiveness’ followed by ‘significant and clinically relevant improvements’ appears contradictory on first reading

At line 443, you mention that “six” of the devices were used across care settings, but it appears from table 4 that a mean of 2 devices were used across the two settings. Please revise the wording as appropriate.

Line 448 - Please provide justification for how results can be considered of clinical importance if ‘clinically important difference has not been established for this version of the SPPB’

Line 493 - Please revise the sentence ‘there was no statistically significant difference in important participant-reported outcomes such as health related quality of life; however, all these outcomes were favourable’ to clarify your findings / make explicit whether significant differences were seen or not

Please add a short section on the implications and next steps for research, clinical practice, and/or public policy between your limitations and conclusion summary 

Please include paragraph numbers (instead of page numbers) in your CONSORT statement

Please remove files Manuscript under review_1 and Manuscript under review_2

Comments from Reviewers:

Reviewer #1: The authors have appropriately responded the all the reviewers comments. The re-review of the methods and reporting of the trial methods and results are much improved and clearer. This is a high quality trial, with well-reported findings, and clinically relevant findings to the rehabilitation field. I have recommended acceptance for publication on that basis. Interestingly, the reviewers have all highlighted that a much more detailed economic analysis will be presented separately which I will very much look forward to seeing. 

The only comment I have remaining is regards to a comment regarding balance between arms after randomisation. I take the authors point that baseline statistical testing is not warranted and opposed by CONSORT (and randomisation has been reported and conducted appropriately). The question I had remaining was whether, from a clinical perspective whether the authors felt there could be potentially any influencing participant factors (if any), that would they would view as clinically relevant. Given that some imbalance occurs by chance, whether any these factors should or could be explored further perhaps in a post-hoc analyses as covariate adjustments to explore the influence on the results. Altman's 1985 paper has a few examples of exploring covariate adjustment principles by strength of prognostic factors but these are mostly decisions guided by clinical expertise and experience: "Comparability of Randomised Groups." Journal of the Royal Statistical Society. Series D (The Statistician), vol. 34, no. 1, 1985, pp. 125-136. JSTOR, www.jstor.org/stable/2987510. It would be useful to get the authors view on this. 

Reviewer #2: The authors have done an excellent job responding to the comments. The manuscript is substantially improved and a stronger communication of the methods and results of this important trial. I only have a few mainly minor suggestions for the authors to consider:

Reviewer 1 

Point 3 response: Instead of 'with allocation concealment' it would be more informative to state the method, such as 'assigned using a remotely prepared web-based randomisation system to…'

Point 19 response: The extra detail in the response about clinician assessed capacity for improvement is informative. It means that a pragmatic assessment was used, incorporating clinical judgement. I think clarification of this detail would be a useful addition to the paper and would aid generalisability. 

Point 27 response: As the selected 15% difference was selected based on experience of what would be clinically important, it would be useful background to explain this in the sample size section. The relationship between the 15% clinically meaningful difference used in the sample size calculation and the effect sizes observed in the SPPB would be a useful point for the discussion to aid interpretation of the clinical importance. 

Author summary: are the +/- SDs?; the 12 point SPPB results are in the what this study adds section, rather than the primary outcome (0-3 continuous SPPB)- would suggest the primary outcome data would be more appropriate.

[LINK]

---

## [Editor Report · Decision Letter 2]

17 Dec 2019

Dear Dr. Hassett,

Thank you very much for re-submitting your manuscript "Digitally-enabled aged care and neurological rehabilitation to enhance outcomes with Activity and MObility UsiNg Technology (AMOUNT) in Australia: a randomised controlled trial." (PMEDICINE-D-19-02056R2) for review by PLOS Medicine.

I have discussed the paper with my colleagues and I am pleased to say that, provided the remaining editorial and production issues are dealt with, we are planning to accept the paper for publication in the journal.

The remaining issues that need to be addressed are listed at the end of this email. Please take these into account before resubmitting your manuscript:

[LINK]

We look forward to receiving the revised manuscript by Dec 20 2019 11:59PM. 

Sincerely,

Louise Gaynor-Brook, MBBS PhD

Associate Editor 

PLOS Medicine

plosmedicine.org

Requests from Editors:

Thank you for adding to your revised manuscript information regarding the cost of the intervention, which we feel has improved the manuscript. 

We note the addition of a new author. PLOS acts according to COPE guidelines, and so requires that any changes to the author listing on a manuscript are confirmed and agreed by all authors. All authors must fulfill the ICMJE criteria for authorship, and meet the CRediT guidelines for contributions, as described here: http://journals.plos.org/plosmedicine/s/authorship.

Please could you, as corresponding author, contact all authors - including the authors being added - to obtain their emailed confirmation that they agree to this change and the contributions made by the proposed new author? You should then forward all the confirmations to us in one file.

Thank you for confirming that access to de-identified data is still being considered by the lead ethics committee, and for suggesting an alternative contact (not an author) in the event that permission is not granted by the ethics committee. We will not be able to publish this work until this issue resolved. 

We note that some of the modifications to the intervention protocol have been summarised in Table 1. Please provide the three versions published during the trial in the supplementary files.

You mention at line 479-481 that not finding “statistically significant difference in important participant-reported outcomes … may reflect low statistical power to demonstrate significance”. Please add this is as a limitation to the final sentence of your Abstract: Methods and Findings.

Line 174 - please specify the cities in which the hospitals were located (as you have in your Abstract) 

Line 231 – please clarify whether a specific amount of time spent upright was required in order to record ‘upright time’ as an outcome 

Please provide references to published questionnaires used in the study. 

Line 429 - please revise to "Six of the devices were used across both inpatient and post-hospital care settings"

Line 448 – please replace ‘favourable’ with another term that clarifies the meaning of the sentence ‘Our results are more favourable than those of the Cochrane systematic review of virtual reality interventions in people after stroke’.

We note your response to our previous request to ‘Please provide justification for how results can be considered of clinical importance if ‘clinically important difference has not been established for this version of the SPPB’ Your explanation is still not clear; please clarify what exactly is meant by the comparison of your results with “the amount of deterioration over a 1- year period in older women living in the community with disability”, when this compares improvements in your primary mobility measure with worsening of mobility through deterioration. 0.2 points at 6 months compared to 0.15 points at 1 year does not appear convincingly clinically important to a non-expert reader. 

Please indicate the relevant sections (e.g. Methods, etc) and paragraph numbers (instead of page numbers) in your CONSORT statement. As an online journal, papers published in PLOS Medicine do not have page numbers.

---

## [Editor Report · Decision Letter 3]

22 Jan 2020

Dear Dr Hassett, 

On behalf of my colleagues and the academic editor, Dr. Christelle Nguyen, I am delighted to inform you that your manuscript entitled "Digitally-enabled aged care and neurological rehabilitation to enhance outcomes with Activity and MObility UsiNg Technology (AMOUNT) in Australia: a randomised controlled trial." (PMEDICINE-D-19-02056R3) has been accepted for publication in PLOS Medicine. 

PRODUCTION PROCESS

PRESS

PROFILE INFORMATION

Thank you again for submitting the manuscript to PLOS Medicine. We look forward to publishing it. 

Best wishes, 

Louise Gaynor-Brook, MBBS PhD

Associate Editor 

PLOS Medicine

plosmedicine.org